# Comparison of the Rostral Epidural Rete Mirabile and the Patterns of Its Blood Supply in Selected Suiformes and Hippopotamuses

**DOI:** 10.3390/ani13040644

**Published:** 2023-02-13

**Authors:** Maciej Zdun

**Affiliations:** 1Department of Basic and Preclinical Sciences, Nicolaus Copernicus University in Torun, Lwowska 1, 87-100 Torun, Poland; maciejzdun@umk.pl; Tel.: +48-56-611-30-93; 2Department of Animal Anatomy, Poznan University of Life Sciences, Wojska Polskiego 71C, 60-625 Poznań, Poland

**Keywords:** anatomy, angiology, artery, encephalon, RERM, vascularization

## Abstract

**Simple Summary:**

The rostral epidural rete mirabile is built of considerable small arterial vessels that anastomose with each other at the base of the cranial cavity. This structure is formed as a result of the division of a large arterial vessel into many small arteries. Furthermore, on the other side of the rete mirabile, these small arteries are joined together to form one large artery, through which blood flows out of the rete mirabile. This system participates in decreasing the temperature of the brain and thus protects the body from thermal stress. Moreover, it influences body–water balance and has a crucial role in the retrograde transfer of neuropeptides. According to old systematics, pigs and hippopotamuses are in one suborder. Hence it was decided to compare their structures and blood supplies. It was found that the shape of the analyzed structure and the inflows differed in these groups.

**Abstract:**

The rostral epidural rete mirabile (*rete mirabile epidurale rostrale*) is built of considerable small arterial vessels that anastomose with each other. This structure is formed as a result of the division of a large arterial vessel into many small arteries. Furthermore, on the other side of the rete mirabile, these small arteries are joined together to form one large artery, through which blood flows out of the rete mirabile. This system participates in decreasing the temperature of the brain and thus protects the body from thermal stress. Moreover, it influences body–water balance and has a crucial role in the retrograde transfer of neuropeptides. The goal of this study was to describe the rostral epidural rete mirabile and pathways that provide blood to this structure as well as compare it in selected Suiformes and hippopotamuses. The study was performed on desert warthogs (*Phacochoerus aethiopicus*), Eurasian wild boars (*Sus scrofa*), collared peccaries (*Pecari tajacu*), pygmy hippopotamuses (*Choeropsis liberiensis*), and common hippopotamuses (*Hippopotamus amphibius*). Preparations were made using the latex method and corrosion cast. An elongated shape characterizes its anatomy with a much wider rostral part than caudal part in the Eurasian wild boars, desert warthogs, and collared peccaries. The main source of blood was the branch to the rostral epidural rete mirabile branched off from the internal carotid artery. Moreover, blood enters the rete by the caudal branch and rostral branch to the rostral epidural rete mirabile. In hippopotamuses, the major source of blood was the rostral branches to the rostral epidural rete mirabile.

## 1. Introduction

The rostral epidural rete mirabile (*rete mirabile epidurale rostrale*) is an arterial system built of considerable small vessels that anastomose with each other. It is situated at the base of the cranial cavity [1,2,3,4,5]. The rostral epidural rete mirabile is found in the cerebral base of Artiodactyla. It is a characteristic structure for this order [6]. The rostral epidural rete mirabile has not been found in representatives of other orders. Mentions of the occurrence of this rete mirabile in humans appear in the works of Leonardo da Vinci. In 1664, Thomas Willis denied these claims and stated that there is no rostral epidural rete mirabile in humans and horses [7]. A detailed analysis of the rete mirabile was made in the dromedary and llama from the Tylopoda suborder. In this species, the rostral epidural rete mirabile has a spongy appearance and is composed of a tight network of anastomosed arteries occupying the whole cavity of the cavernous sinus. It is H-shaped and composed of two lobes on each side of the hypophysis. Each lobe is divided into rostral and caudal parts to the hypophysis. The rostral part represents a major part of the rete mirabile. The shape of the unilateral structure was described as an elongated oval [4,8]. Similar observations have been made in some representatives of Bovidae and Cervidae from the Ruminantia suborder. In these animals, the rete mirabile assumes an analogous shape and is asymmetrical in the rostral part. The caudal part is acute. Bilateral structures are connected to each other rostrally and caudally to the hypophyseal gland [2,9]. In histological studies conducted on sheep, it was found that the vessels of the rete mirabile and the cavernous sinus share a common adventitia [10]. In addition, there are fenestrations in the arterial vessels of the rete. The diameter of these fenestrations depends on the phase of the cycle [11]. A detailed list of species in which this rete has been found, and information about the vessels that connect to it are included in the discussion section. In addition to the rostral epidural rete mirabile, other rete mirabiles have also been described. In the limbs of the northern tamandua (*Tamandua mexicana*), the rete mirabile was found. The superficial brachial (*arteria brachialis superficialis*), cubital transverse (*arteria transversa cubiti*), and cranial and caudal interosseous arteries (*arteria interossea cranialis et caudalis*) have a structure of rete mirabile sending vessels to the muscles at the craniolateral part of the forearm. The medial and lateral branches of the superficial antebrachial artery (*arteria antebrachialis superficialis*), which arise as an extension of the superficial brachial artery, also form the rete mirabile. This structure is flat and wide. The major vessels of the forearm in the southern tamandua (*T. tetradactyla*), Cyclopes, and sloths also form the rete mirabile [12,13]. The anatomy of the rostral epidural rete mirabile is closely determining its function. This structure participates in the selective decreasing of the brains’ temperature and thus protects the body from thermal stress [14,15]. Moreover, it influences body–water balance and plays an important role in the retrograde transfer of neuropeptides such as progesterone and β-endorphin [16]. This mechanism is made possible by the location of the rostral epidural rete mirabile in the cavernous sinus (*sinus cavernosus*). Arterial vessels contain blood at a higher temperature. The blood returning from the nasal cavity with lower temperature contributes to cooling arterial blood located in the rostral epidural rete mirabile. Due to the close proximity of the vessels of the rete mirabile and the cavernous sinus, there is a transfer of heat from the arterial blood to the cooler venous blood. Cooled arterial blood follows to the vessels supplying the brain [14]. As a consequence, it affects the welfare of an animal. Because of the physiological implications of the anatomical structure and location of the rostral epidural rete mirabile, it seems essential and necessary to describe its characteristics.

Three suborders were separated according to old systematics based on morphological features in the Artiodactyla order. These were Suiformes, Ruminantia, and Tylopoda [17,18]. According to more recent studies using genetic studies, Whippomorpha were further separated. This is because hippopotamuses have been shown to be more closely related to Cetacea than pigs [19,20]. According to old systematics, hippopotamuses belonged to the Suiformes, while more recent studies classified them as Whippomorpha. Among the analyzed species were those occurring quite commonly as well as relatively rare. 

The information provided in this study, which focuses on anatomy, can be helpful to scientists, especially physiologists. In addition, information about this topic may be helpful to veterinarians who care for animals in zoos or animal rescue centers. Blood supply pathways to the brain are essential during clinical procedures only in cases of congestion or syncope due to poor brain perfusion. The brain is vulnerable to hypoxia and cannot accumulate glucose or glycogen, so an adequate blood supply is critical [21,22].

The aim of this study is to describe the rostral epidural rete mirabile and pathways that provide blood to the encephalon through the rete mirabile in selected Suiformes and hippopotamuses. A further purpose is also to compare the form of this structure and arterial patterns of this rete mirabile between the mentioned groups.

## 2. Materials and Methods

### 2.1. Animals

The study was performed on five desert warthogs (*Phacochoerus aethiopicus*), ten Eurasian wild boars (*Sus scrofa*) from the Suidae family, six collared peccaries (*Pecari tajacu*) from the Tayassuidae family from the Suiformes, two pygmy hippopotamuses (*Choeropsis liberiensis*), and three common hippopotamuses (*Hippopotamus amphibius*). The analyzed cadavers were obtained as post-mortem material from private breeders and zoos in Poland or obtained through hunting. The animals had not been euthanized or hunted for an experimental purpose. Of the cadavers obtained from hunting, only animals without damage to the head and neck were included in the study. Both genders were used in the study. Approval from research ethics committees was not needed to accomplish the aims of this study because the experimental work was conducted only on dead specimens. All procedures involving cadavers in accordance with the law on 15 January 2015 for the protection of animals used for scientific or educational purposes did not require the approval of the local ethics committee (Journal of Laws 2015, item 266).

### 2.2. Methods

In the study, the classical techniques used in cardiovascular anatomy research were used. The specimens were prepared in two ways and assigned to one of the methods in random order. In twenty-one specimens (Table 1), injection of both common carotid arteries with a red solution of the chemo-setting acrylic material Duracryl^®^ Plus (SpofaDental, Jicín, Czech Republic) was performed. In addition, in four specimens (Table 1), a blue solution of the chemo-setting acrylic material Duracryl^®^ Plus (SpofaDental, Jicín, Czech Republic) was injected into the bilateral external jugular vein (*vena jugularis externa*). When the injected solution hardened, the specimens were moved to a special container with a washing powder (Persil, Germany) solution for enzymatic maceration. The temperature used in this process was 40 °C. The maceration lasted 45 days, during which the specimens were controlled every seven days. An arterial casting of the head and cranial cavity on a skeletal scaffold were obtained as a final step in the described process. The second method used for five specimens (Table 1) consisted of injecting red liquid LBS 3060 latex into both the right and left common carotid arteries. Furthermore, in two specimens (Table 1), a blue liquid LBS 3060 latex was injected into the bilateral external jugular vein. Later, the specimens were submerged in 5% formalin solution for 10 days, and the specimens were rinsed in water for 48 h prior to the next step to flush out excess formalin solution for safety reasons. Additional protection was provided with a ventilation system in a room that was used for the preparation and protection of the preparation person using latex gloves, a filtered mask, and safety googles. The ventilation system was set for 15 air changes per hour. The skull bone was cut watchfully using an oscillating saw and then opened to visualize the soft tissue and enable preparation. In the next step, the blood vessels were manually prepared using surgical and anatomical instruments. Then, the excess connective tissue was cleaned from the prepared arterial vessels for better visualization. Using this method, the visible blood vessels patterns on the animal’s soft tissues were obtained. The diameter of the casts of blood vessels providing blood to the rete mirabile before the vessel entered the rete mirabile was measured using calipers. Subsequently, all the diameters of a given individual were added together, taking this value as 100%. In the next step, the proportion that a given vessel represents to the sum of all vessels was calculated, and this value was given as a percentage. The average value for a given vessel was calculated from the analyzed specimens in a given family. This presentation of the data was intended to show which vessel has the largest diameter in a given group of animals proportionally.

The names of the anatomical structures were standardized according to the *Nomina Anatomica Veterinaria* [23].

## 3. Results

### 3.1. Suidae Family

The rostral epidural rete mirabile is supplied through several ways. They all originate from the common carotid artery (*arteria carotis communis*). This vessel is divided into the external carotid artery (*arteria carotis externa*) and the internal carotid artery (*arteria carotis interna*). The internal carotid artery exists only as an initial segment. It branches off from the common carotid artery together with the condylar artery (*arteria condylaris*) and occipital artery (*arteria occipitalis*). The rest of the extracranial segment of the internal carotid artery is obliterated. The initial part of this artery becomes the ramus to the rostral epidural rete mirabile (*ramus ad rete mirabile epidurale rostrale*). This is the primary blood source to the rostral epidural rete mirabile. In the representatives of this family, the diameter of this vessel is 84.5% of the total diameter of the arteries supplying blood to the rostral epidural rete mirabile (Figure 1). In the Eurasian wild boars and desert warthogs, the caudal epidural rete mirabile (*rete mirabile epidurale caudale*) is located caudally to the ramus to the rostral epidural rete mirabile. The condylar artery and occipital artery merge with the caudal epidural rete mirabile. This is a paired structure formed from a few vessels resembling a fishing net.

Another source of blood for the rostral epidural rete mirabile is branches of the maxillary artery (*arteria maxillaris*). The maxillary artery is a continuation of the external carotid artery. From the maxillary artery, the middle meningeal artery branches off (*arteria meningea media*). From this vessel, the caudal branch to the rostral epidural rete mirabile (*ramus caudalis ad rete mirabele epidurale rostrale)* (Figure 2) branches off and joins the rete mirabile. In this specimen, this vessel is 7% of the total diameter of the arteries supplying blood to the rete mirabile. Next, the external ophthalmic artery (*arteria ophthalmica externa*) branches off from the maxillary artery. From this, the vessel branches off the rostral branch to the rostral epidural rete mirabile (*ramus rostralis ad rete mirabele epidurale rostrale*). It is a single vessel in the analyzed representatives of the Suiformes suborder. In one Eurasian wild boar, it was double bilaterally. The diameter of this vessel is 8.5% of the total diameter of the arteries supplying the rostral epidural rete mirabile.

The rostral epidural rete mirabile is located at the base of the cranial cavity on the basisphenoid bone laterally surrounding the hypophyseal fossa. This is a well-developed, paired structure composed of numerous tiny anastomosing arteries (Figure 2). An elongated shape characterizes its form with a much wider rostral part than caudal. Bilateral structures are connected by several vessels in the rostral part. In a cross-sectional view, the rostral epidural rete mirabile has an oval outline. All arteries in the rete mirabile are similar in diameter (Figure 3). In the center of the rete mirabile, there is no single vessel with a significantly larger lumen than the others. There was no direct connection between the rostral and caudal epidural retia mirabilia. The intracranial segment of the internal carotid artery emerges from the rostral epidural rete mirabile. It forms the cerebral arterial circle (*circulus arteriosus cerebri*), whose branches supply the encephalon.

Veins at the base of the brain are expressed in the form of venous sinuses. The most important for functional reasons is the cavernous sinus (Figure 4). This sinus has a porous structure due to the numerous small spaces filled with vessels of the arterial rostral epidural rete mirabile. In the central part of this sinus, there is a fragment connecting the bilateral sinuses. The shape of the cavernous sinus is that of a rete mirabile. It is the intercavernous sinus (*sinus intercavernosus*), situated rostrally from the sella turcica.

### 3.2. Tayassuidae Family

In this family, all ways that provide blood to the rostral epidural rete mirabile also take their origin from the common carotid artery. The internal carotid artery branches off from the common carotid artery together with the condylar artery and occipital artery (Figure 5). The main extracranial part of this vessel is obliterated. Blood enters the rete mirabile by the ramus to the rostral epidural rete mirabile. This is the primary blood source to the rostral epidural rete mirabile. The diameter of this vessel is 66% of the total diameter of the arteries supplying blood to the rostral epidural rete mirabile. In the collared peccaries, the caudal epidural rete mirabile does not occur.

Next, the caudal branch to the rostral epidural rete mirabile branches off from the middle meningeal artery. The diameter of this vessel is 17% of the total diameter of the arteries supplying blood to the rostral epidural rete mirabile. More rostrally, the rostral branch to the rostral epidural rete mirabile arises (Figure 6). It is a branch of the external ophthalmic artery. It is a single vessel. In two collared peccaries, it was double or triple bilaterally. The diameter of this vessel is 17% of the total diameter of the arteries supplying the rostral epidural rete mirabile. 

The position of the rete mirabile is analogous to that in the representatives of the Suidae family (Figure 7). In this species, it is also a well-developed, paired structure composed of numerous tiny anastomosing arteries. An elongated shape characterizes its form with a much wider rostral part than caudal. The rostro-lateral part is well developed, which makes the bilateral retes resemble the appearance of a butterfly. Bilateral structures are connected by several vessels in the rostral part. In a cross-section view, the rete mirabile looks like one in the Suidae family. The rostral epidural rete mirabile is also located in the cavernous sinus. The structure of this sinus did not differ from that described in the Suidae family. In shape, it corresponded to the rete mirabile, closely adhering to the vessels of the rete.

### 3.3. Hippopotamuses

The internal carotid artery arises from the common carotid artery. From the initial part of this vessel, the ramus to the rostral epidural rete mirabile arises (Figure 8). This is a double vessel. Close to the point where it enters the rostral epidural rete mirabile, it sometimes occurs as a single vessel. The diameter of this vessel is 29% of the total diameter of the arteries supplying blood to the rostral epidural rete mirabile. Another feature that occurs in hippopotamuses is the presence of the caudal epidural rete mirabile along the course of this vessel. It is a paired structure formed from few vessels. 

From the maxillary artery, the middle meningeal artery arises, from which the caudal branch to the rostral epidural rete mirabile branches off (Figure 9). The diameter of this vessel is 10% of the total diameter of the arteries supplying blood to the rostral epidural rete mirabile. Next, the external ophthalmic artery arises from the maxillary artery. From this vessel, the rostral branches to the rostral epidural rete mirabile branch off. The rostral branches to the rostral epidural rete mirabile are present in numbers three to five. In three cases, bilaterally and one unilaterally, it is a triple vessel. In one case, bilaterally, it is quadruple. In one case, unilaterally, this vessel is in the number of five. The total diameter of these branches is 61% of the total diameter of the arteries supplying blood to the rostral epidural rete mirabile. Most blood in this species enters the rostral epidural rete mirabile through these vessels.

The rostral epidural rete mirabile is a well-developed, paired structure composed of numerous tiny anastomosing arteries. No widening is observed in the rostral part, but it has a similar diameter over its entire length. Bilateral structures are connected by several vessels in the caudal part.

## 4. Discussion

The presence of the rostral epidural rete mirabile is a feature of almost all animals from the Artiodactyla order. Its occurrence has been found in the Tylopoda suborder, which includes the camels of the Old and New Worlds, specifically the dromedary (*Camelus dromedarius*) and Bactrian camel (*Camelus bactrianus*) as well as the llama (*Lama glama*) and guanaco (*Lama guanicoe*) [4,24,25,26]. In the Ruminantia suborder, this structure has been found in all described animals from infraorder Pecora, e.g., giraffe (*Giraffa camelopardalis*), sitatunga (*Tragelaphus spekei*), nyala (*Tragelaphus angasi*), greater kudu (*Tragelaphus strepsiceros*), common eland (*Taurotragus oryx*), nilgai (*Boselaphus tragocamelus*), springbuck (*Antidorcas marsupialis*), blackbuck (*Antilope cervicapra*), dik-dik (*Madoqua kirkii*), saiga (*Saiga tatarica*), oribi (*Ourebia ourebi*), domestic cattle (*Bos taurus taurus*), goat (*Capra hircus*), domestic sheep (*Ovis aries*), banteng (*Bos javanicus*), yak (*Bos mutus f. grunniens*), European bison (*Bison bonasus*), American bison (*Bison bison*), common wildebeest (*Connochaetes taurinus*), mouflon (*Ovis aries musimon*), chital (*Axis axis*), European red deer (*Cervus elaphus*), wapiti (*Cervus elaphus Canadensis*), milu (*Elaphurus davidianus*), Eld’s deer (*Rucervus eldii*), sika deer (*Cervus nippon*), fallow deer (*Dama dama*), moose (*Alces alces*), roe deer (*Capreolus capreolus*), reindeer (*Rangifer tarandus*), and Chinese muntjac (*Muntiacus reevesi*) [27,28,29,30,31,32,33,34,35,36,37,38]. In contrast, in Tragulidae of the Tragulina infraorder, the internal carotid artery anastomoses directly to the cerebral arterial circle due to the lack of the rostral epidural rete mirabile. This group of animals survived without significant changes from the Miocene when they separated from the stem of the phylogenetic tree of the Artiodactyla order. It is not known whether the lack of the rostral epidural rete mirabile is an apomorphic or plesiomorphic trait, that is, whether it was lost as a result of evolution or never developed in this group of animals. This is the only group of animals among the Artiodactyla in which the occurrence of this structure has not been described [39,40]. In Suiformes, a detailed analysis of the anatomy of this structure has been described in the domestic pig (*Sus scrofa f. domestica*) [3,41,42].

During the analysis of the anatomy of the rostral epidural rete mirabile in the cross-section, it can be seen that two types of rete mirabiles can be identified. A rete mirabile formed from vessels of similar diameter has been identified as a bipolar type [1]. The same type of rete mirabile has been found in the analyzed species i.e., in the desert warthogs, Eurasian wild boars, collared peccaries, pygmy hippopotamuses, common hippopotamuses, and in the common wildebeests, sheep and goats [2,5]. In domestic cattle, European bison, red deer, roe deer, and fallow deer, one vessel has a much larger diameter than the other vessels in the rete mirabile [2]. According to Simoens’ [1] classification, this rete mirabile was called the fascicular type.

In all the analyzed representatives of Suiformes, the rostral branch to the rostral epidural rete mirabile is a single, weak vessel. In the hippopotamuses, the vessels are more numerous (from three to five). They are also the primary source of blood for the rete mirabile (61% of the total diameter of all vessels which provide blood to the rete compared to 17% in Suidae and 8.5% in collared peccaries). In the described Artiodactyla, those branches are developed in various ways. In the Tylopoda, these branches are very well developed. They occur in several numbers, and, close to the point of origin, they divide secondarily into a more significant number of small vessels, thus contributing to the expansion of the volume of the rete mirabile on the rostral side [8,24]. The rostral portion of the rete mirabile is built similarly in the common wildebeest from the Ruminantia [5]. In some representatives of the Ruminantia, such as domestic goats, domestic sheep, cattle, European bison, roe deer, red deer, and fallow deer, those branches were described in a number of two to four. However, they were not such strongly developed as in the Tylopoda or common wildebeest. Moreover, in the Tylopoda and Ruminantia, rostral branches to the rostral epidural rete mirabile branch off mainly from the maxillary artery and less often from the external ophthalmic artery [2,5,8,24]. In the analyzed species, all rostral branches to the rostral epidural rete mirabile branch off from the external ophthalmic artery. The caudal branch to the rostral epidural rete mirabile in all analyzed species in this study is a vessel with a very small lumen. It branches off from the middle meningeal artery. In the llama, this vessel is also described as an artery with a small lumen but branches off directly from the maxillary artery [8]. In the camel, it has a larger diameter [24]. In the Bovini and Cervidae, the diameter is comparable to the diameter of each rostral branch to the rete mirabile. A much larger diameter is found in the domestic goat, domestic sheep, and common wildebeest [2,5]. The phenomenon of obliteration of the extracranial part of the internal carotid artery has been described in the domestic pig and Eurasian wild boar [43]. According to this information, the vessel connecting to the rete mirabile from its caudal side has named the branch to the rostral epidural rete mirabile. This vessel that originated from the initial part of the internal carotid artery is the artery with the largest diameter in the desert warthogs and Eurasian wild boars. Such observations have also been made in the domestic pig [3,27,43].

The caudal epidural rete mirabile as the paired structure is found in desert warthogs, Eurasian wild boars, and hippopotamuses. This structure is also described in domestic pigs [3,27]. Unpaired caudal epidural rete mirabile has been noticed in the representatives of the Bovini tribe, such as domestic cattle, banteng, yak, American bison, and European bison [36]. In contrast to animals of the Suiformes, this rete mirabile in the members of the Bovini tribe is directly connected to the rostral epidural rete mirabile. The connection between these structures is also found in hippopotamuses in this research.

Physiologically, the encephalon exhibits a relatively large temperature fluctuation of 2–4 °C, which is associated with the naturally varied activities of life. In addition to the observed physiological hyperthermia, some situations cause the temperature of the encephalon to rise above acceptable norms. This can cause numerous abnormalities leading to negative effects affecting neural activity and function [44]. For this reason, in most mammals, we observe various adaptations in the course and structure of blood vessels to protect the brain from the negative effects of high temperatures [45]. An example of such adaptations is the presence of a well-developed rostral epidural rete mirabile, as observed in the Artiodactyla. The first description of the functional significance of the presence of a rostral epidural rete mirabile located close to the base of the brain and the relationship of its structure to its role in selective brain cooling was described in 1969 by Hayward and Baker [46]. This mechanism allows the animal’s brain temperature to be maintained at 40.5 °C during muscular exertion. In the analyzed species, the rete mirabile is a well-developed structure located in the cavernous sinus. There were several experiments conducted on different species that showed the temperature of the encephalon is lower than the temperature of the blood in the common carotid artery. The difference ranged from 1.3 °C in domestic cattle up to 3.98 °C in camels [47,48,49]. Therefore, it can be assumed that it is an adaptation to this function. The well-developed structure has a large contact area with the cavernous sinus, which may result in a more efficient heat exchange between the arterial blood in the rete mirabile and the venous blood in the cavernous sinus. The rostral epidural rete mirabile as an evolutionary achievement has provided Artiodactyla with a better opportunity to adapt in terms of physiology to changing environmental conditions throughout history [14,39]. According to Strauss et al. [14], the occurrence of this structure in hippopotamuses was questionable due to the lack of data in this area. The role of this phenomenon is not limited to protecting the encephalon from overheating. It also affects the body’s water balance. It represents an adaptation to extremely hot and dry environments [15]. This is because it allows for a reduction in evaporation. In Dorper sheep, it was found to possibly save even more than 2.5 liters of water per day in an individual weighing about 50 kg [50]. Another role of the described structure can be found in the literature. The mechanism of neuropeptide transport (Lh-Rh, progesterone, and β-endorphin) in the cavernous sinus and the rostral epidural rete mirabile was described in 1992 by Krzymowski et al [16]. In the cavernous sinus, the concurrent exchange of oxytocin, gonadoliberin [51,52], and dopamine takes place [53]. The limited mass of the hormones mentioned above, which varies from 0.19 kDa to 3.4 kDa, enables this process to be performed. Other substances such as prolactin and luteinizing hormone have no possibility to exchange this way, most likely due to the higher mass of these hormones, i.e., 23 kDa prolactin and 30 kDa luteinizing hormone, [54]. Another substance physiologically important for this area is carbon monoxide (CO). The light rays due to the phototransduction mechanism enable its synthesis on the retina [55]. CO may influence various areas of the brain and its activity through the concurrent exchange from ophthalmic region circulation via the cavernous sinus with CO from the nearby veins to the rostral epidural rete mirabile.

The function of the rete mirabile mentioned above indicates how multi-dimensional the physiological importance of the anatomical structure is for Artiodactyla species’ biology. 

## 5. Conclusions

This study provides information about the anatomy of the rostral epidural rete mirabile and the patterns of its blood supply in desert warthogs, Eurasian wild boars, collared peccaries, pygmy hippopotamuses, and common hippopotamuses. It is an elongated shape with a much wider rostral part in Suiformes. In hippopotamuses, no such widening was found. The caudal epidural rete mirabile is absent only in the collared peccaries. In all examined species, the rete mirabile is a bipolar type. The main blood vessel supporting the rete mirabile is the branch to the rostral epidural rete mirabile from the internal carotid artery. Other sources of blood contributed negligibly to its supply in Suiformes. In hippopotamuses, the main source of blood is the rostral branches to the rostral epidural rete mirabile. The rostral epidural rete mirabile as an evolutionary achievement has provided Artiodactyla with a better opportunity to adapt in terms of physiology to changing environmental conditions throughout history. The above research adds to the state of knowledge on the occurrence of this structure in hippopotamuses, in which it was not clear until now whether this structure exists. This could be useful information in the discussion of the evolution of this structure in Artiodactyla.

## Figures and Tables

**Figure 1 animals-13-00644-f001:**
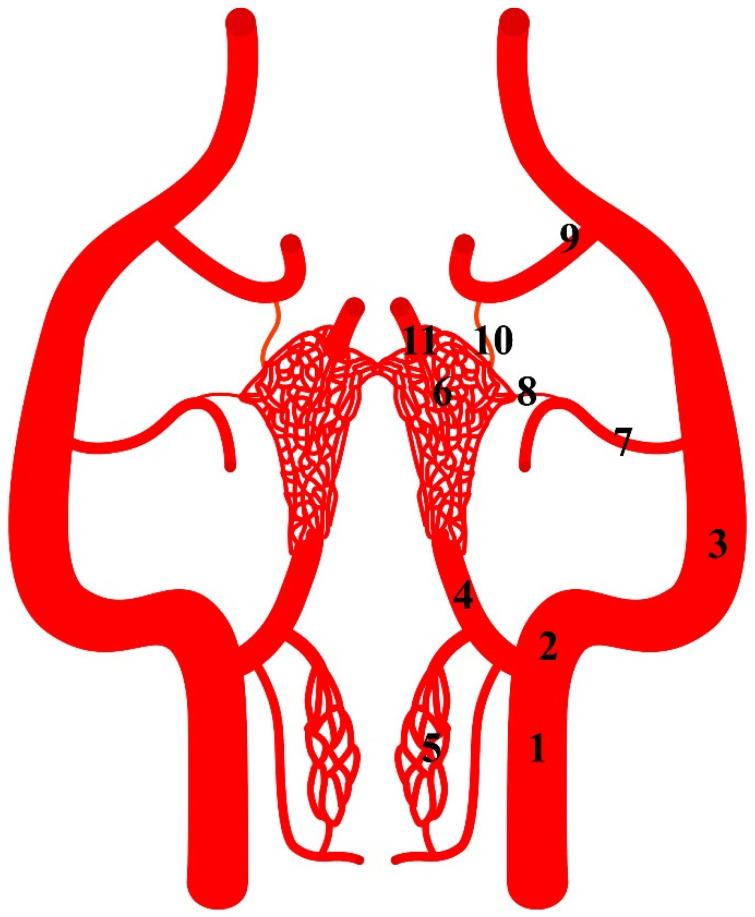
Diagram of the arterial pattern in the Eurasian wild boar (*Sus scrofa*). 1—common carotid artery; 2—external carotid artery; 3—maxillary artery; 4—branch to the rostral epidural rete mirabile; 5—caudal epidural rete mirabile; 6—rostral epidural rete mirabile; 7—middle meningeal artery; 8—caudal branch to the rostral epidural rete mirabile; 9—external ophthalmic artery; 10—rostral branch to the rostral epidural rete mirabile; 11—intracranial segment of the internal carotid artery.

**Figure 2 animals-13-00644-f002:**
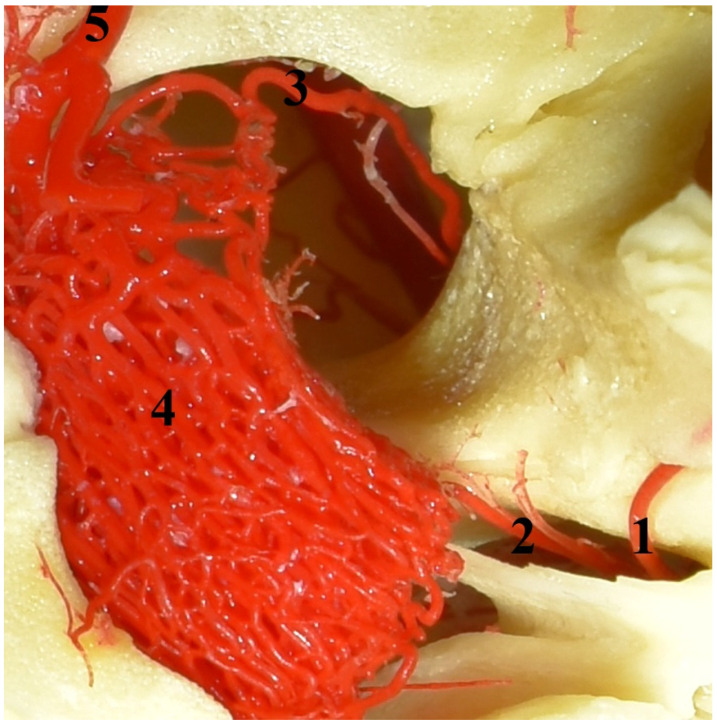
The rostral epidural rete mirabile in the Eurasian wild boar (*Sus scrofa*). A dorsal view and corrosion cast. 1—caudal branch to the rostral epidural rete mirabile; 2—middle meningeal artery; 3—rostral branch to the rostral epidural rete mirabile; 4—rostral epidural rete mirabile; 5—intracranial part of the internal carotid artery.

**Figure 3 animals-13-00644-f003:**
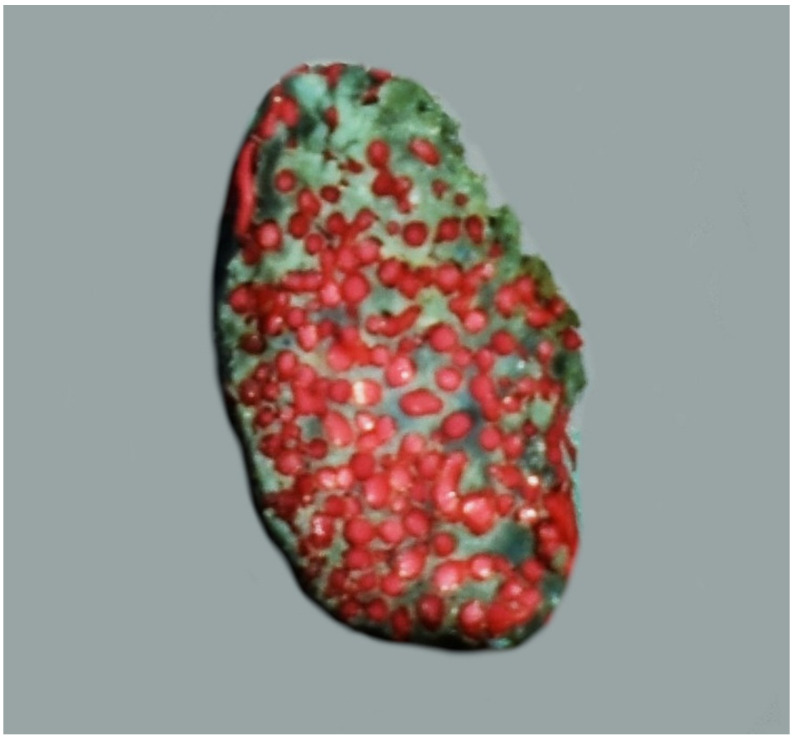
The cross-sectional view of the rostral epidural rete mirabile in the desert warthogs (*Phacochoerus aethiopicus*). A latex cast. Red—arteries of the rete mirabile; blue—cavernous sinus.

**Figure 4 animals-13-00644-f004:**
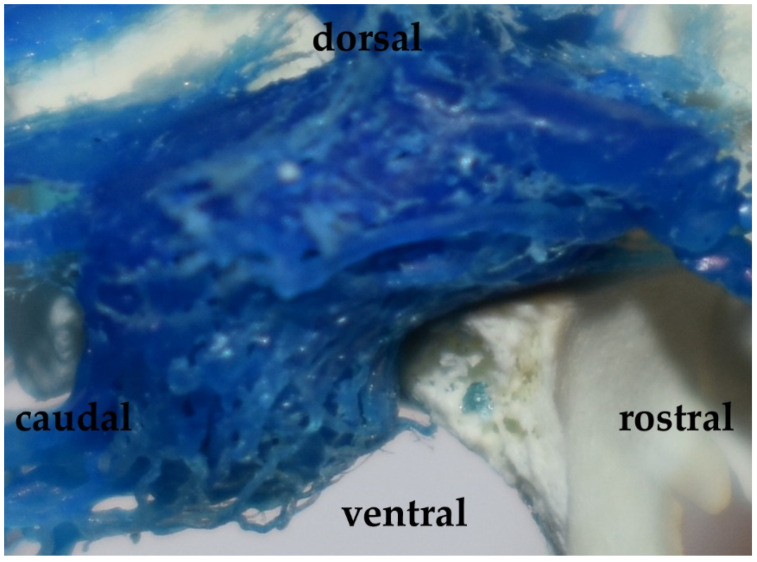
The cavernous sinus in the Eurasian wild boar (*Sus scrofa*). A lateral view and corrosion cast.

**Figure 5 animals-13-00644-f005:**
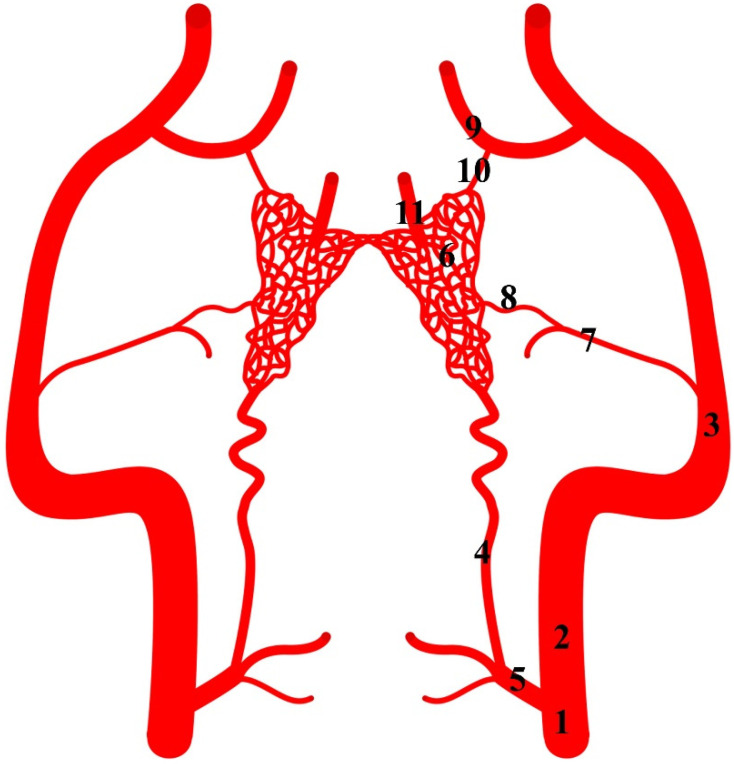
Diagram of the arterial pattern in the collared peccary (*Pecari tajacu*). 1—common carotid artery; 2—external carotid artery; 3—maxillary artery; 4—branch to the rostral epidural rete mirabile; 5—initial part of the internal carotid artery; 6—rostral epidural rete mirabile; 7—middle meningeal artery; 8—caudal branch to the rostral epidural rete mirabile; 9—external ophthalmic artery; 10—rostral branch to the rostral epidural rete mirabile; 11—intracranial segment of the internal carotid artery.

**Figure 6 animals-13-00644-f006:**
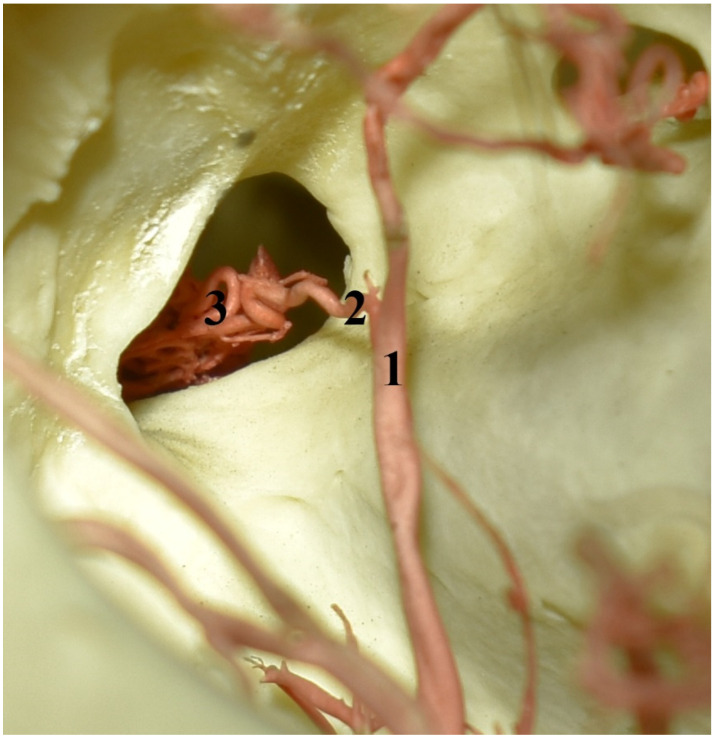
The eye socket of the collared peccary (*Pecari tajacu*). A caudolateral view and corrosion cast. 1—external ophthalmic artery; 2—rostral branch to the rostral epidural rete mirabile; 3—rostral epidural rete mirabile.

**Figure 7 animals-13-00644-f007:**
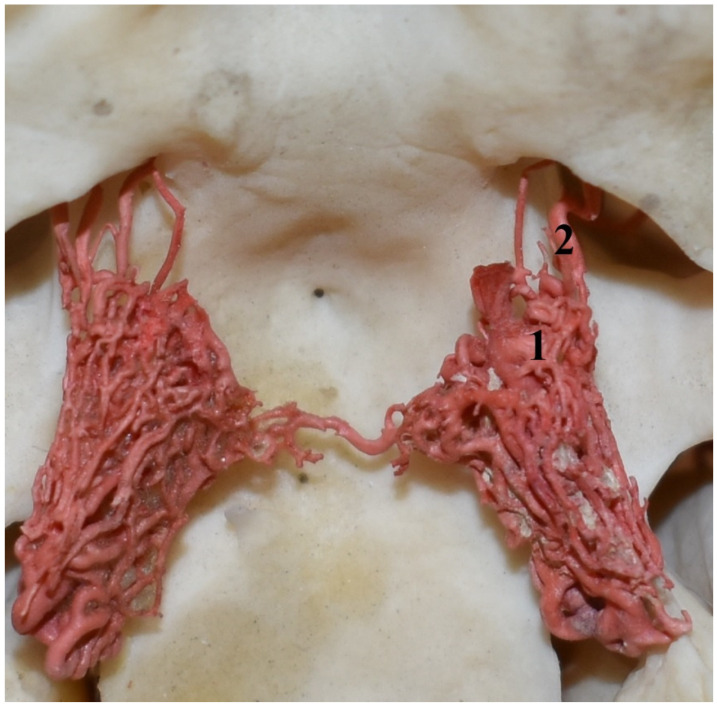
Rostral epidural rete mirabile in the collared peccary (*Pecari tajacu*). A dorsal view and corrosion cast. 1—rostral epidural rete mirabile; 2—rostral branches to the rostral epidural rete mirabile.

**Figure 8 animals-13-00644-f008:**
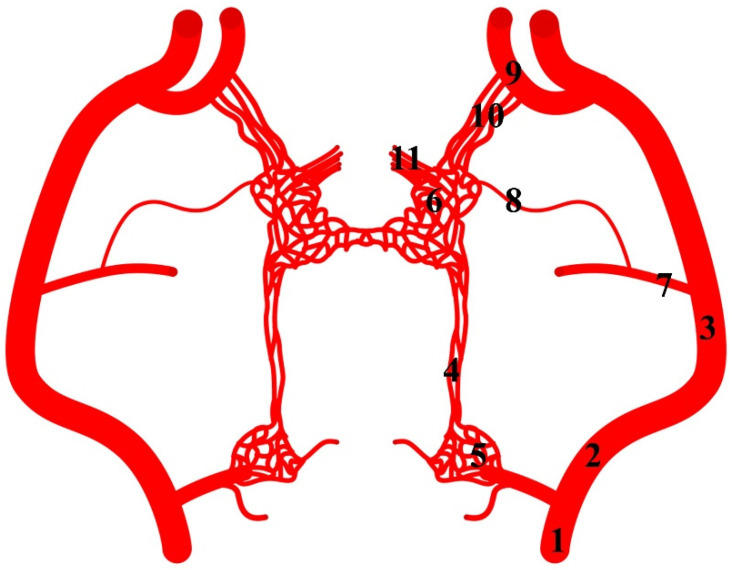
Diagram of the arterial pattern in the common hippopotamus. 1—common carotid artery; 2—external carotid artery; 3—maxillary artery; 4—branch to the rostral epidural rete mirabile; 5—caudal epidural rete mirabile; 6—rostral epidural rete mirabile; 7—middle meningeal artery; 8—caudal branch to the rostral epidural rete mirabile; 9—external ophthalmic artery; 10—rostral branches to the rostral epidural rete mirabile; 11—intracranial segment of the internal carotid artery.

**Figure 9 animals-13-00644-f009:**
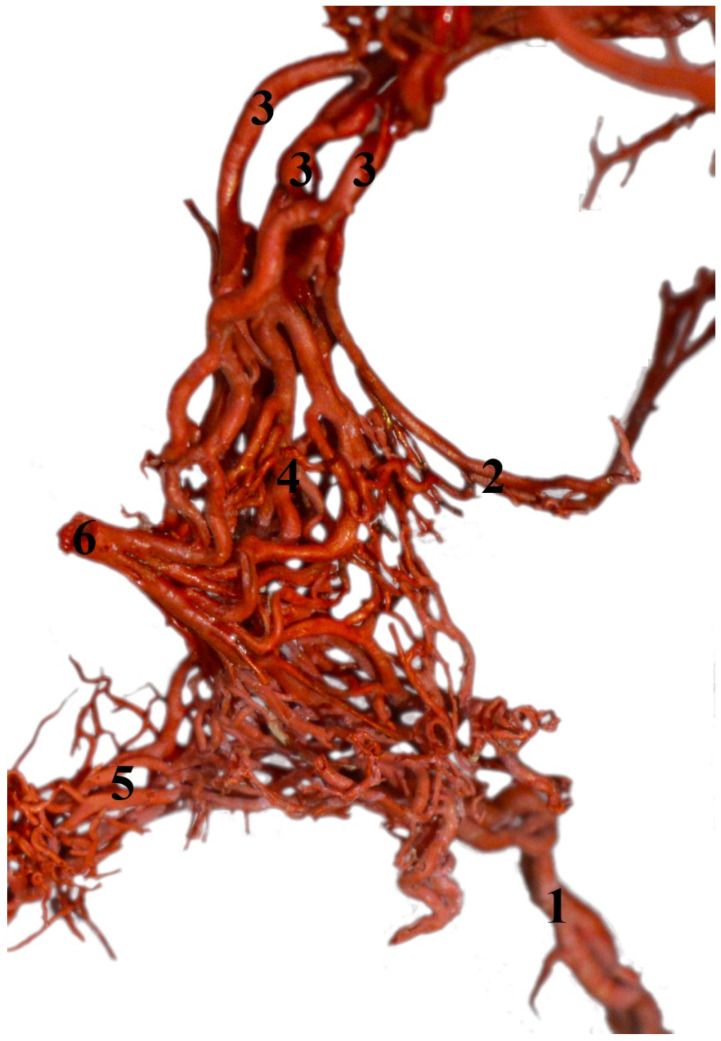
Rostral epidural rete mirabile in the common hippopotamus (*Hippopotamus amphibius*). A dorsolateral view and corrosion cast. 1—branch to the rostral epidural rete mirabile; 2—caudal branch to the rostral epidural rete mirabile; 3—rostral branches to the rostral epidural rete mirabile; 4—rostral epidural rete mirabile; 5—connection between the right and left rete mirabile; 6—intracranial part of the internal carotid artery.

**Table 1 animals-13-00644-t001:** The number of specimens examined in this study.

Species	Liquid into the Common Carotid Artery		Liquid into the External Jugular Vein		Liquid into the Common Carotid Artery and External Jugular Vein	
	Duracryl^®^ Plus (Red)	LBS 3060 Latex (Red)	Duracryl^®^ Plus (Blue)	LBS 3060 Latex (Blue)	Duracryl^®^ Plus (Red + Blue)	LBS 3060 Latex (Red + Blue)
desert warthogs (*Phacochoerus aethiopicus*)	3	1	-	-	1	-
Eurasian wild boars (*Sus scrofa*)	6	1	1	-	1	1
collared peccaries (*Pecari tajacu*)	4	-	-	-	1	1
pygmy hippopotamuses (*Choeropsis liberiensis*)	2	-	-	-	-	-
common hippopotamuses (*Hippopotamus amphibius*)	2	1	-	-	-	-

## Data Availability

The data presented in this study are available on request from the corresponding author.

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
