# Peer review of "Comparison of the Rostral Epidural Rete Mirabile and the Patterns of Its Blood Supply in Selected Suiformes and Hippopotamuses"

_animals, 2023, doi:10.3390/ani13040644_

Round 1

Reviewer 1 Report

Review

This manuscript is a fair attempt to describe and compare rostral epidural rete mirabile in selected Suiformes and Hippopotamuses. However, after reading this manuscript some remarks come to mind.

- line 103-104 authors write: “The second method, used for five specimens, consisted of introducing liquid LBS 3060 latex into the bilateral common carotid arteries” however, the authors did not include figures of vascular specimens filled with latex in the manuscript however, the authors did not include figures of latex-filled vascular specimens in the manuscript, which would undoubtedly have enabled the analysis of the location of the vessels and adjacent brain structures

- line 104-106 authors write that blue latex was injected into the bilateral external jugular  vein. This corresponds to lines 195-200, but that it is not included in any figure. The description of a rather complicated and comparative angiology should be accompanied by appropriate illustrations to support the claims made in the text of the manuscript.

The lack of vascular morphometry does not facilitate the comparison of angiological data. I highly recommend taking these measurements.

Fig 5 - too high magnification of the corrosion specimen weakly correlates with the correct diagrams (Fig: 1, 2, 3).

- line 256-286: physiological content regarding thermoregulation and retrograde transport of neuropeptides, although interesting, have no direct connection with the presented work.

Reviewer 2 Report

Thank you very much for the opportunity to review the manuscript entitled “Comparison of the rostral epidural rete mirabile and the patterns of its blood supply in selected Suiformes and Hippopotamuses”. This is a very interesting work, which could be published in this journal after solving several comments and suggestions included in the annexed word.

Reviewer 3 Report

I suggest a number of major changes that will strengthen the manuscript.

This is an interesting paper that provides new information regarding the anatomical characterization of the rostral epidural rete mirabile (rete mirabile epidurale rostrale), comparing it among selected Suiformes (Desert warthogs, Eurasian wild boars, Collared peccaries) and Hippopotamuses (Pygmy hippopotamuses, and Common hippopotamuses.

The study was based on traditional anatomical techniques, preparing the specimens by two methods:

-        The first one by injecting common carotid arteries with a red solution of the chemo-setting acrylic material Duracryl Plus, and in addition a blue solution of the chemo setting acrylic material Duracryl Plus was injected into the bilateral external jugular vein.

-        The second one introducing liquid LBS 3060 latex into the bilateral common carotid arteries; in addition, a blue liquid LBS 3060 latex was injected into the bilateral external jugular vein.

The methods and techniques used are sound and appropriate for the study and the results are interesting.

The interesting studies on the rostral epidural rete mirabile (rete mirabile epidurale rostrale) comparing data among different mammalian species add new insights not only for neuroanatomy but also for neurosurgery and neurosciences researchers.

However, the manuscript should be substantially revised by addressing the following points:

 Introduction section: pleas avoid to describe details not consistent with the topic of the paper (from line 56 to line 64).One 56 of the species…… concern group [17]”.

 Please, add information on the studies (anatomical or physiological) on the rostral epidural rete mirabile, performed in other mammal’s species including human and describing the anatomical results that researchers found in their studies (not just the list of species).

Results section:

The results section should be organized into separate paragraphs for each species.

The anatomical description and results could be summarized with the use of a table indicating the structure you systematically measure or described and the results in a standardized method and in order to highlight also the peculiar differences between species.

 A scientific paper, cannot contain terms such as those used in the manuscript (see below) to describe the results. Correct the results section with a scientific approach and if possible, a quantitative approach in the comparison between species; both when you describe a structure anatomically and when you evaluate measurements of morphometric parameters (quantitative data).

Line 123, avoid terms as “…is a very strong…”

Line 124, “Especially large lumen…” please elucidate what do you mean. How wide is the lumen? Can you measure it? Can you correlate the size of the lumen with animal size or brain weight or other variables? In an absolute sense, this information for its own sake has little value.

Line 126, avoid terms as “… is not as large in diameter as in the…..”. How large in diameter is it compared to the others?

Line 185, avoid terms as ..”it seems proportionally smaller than…”.

Line189, avoid terms as “….. are similar in diameter…….”. Can you measure it?

Line 196, please describe the cavernous sinus comparing data among species. There are no figures on this structure. Why not? Pleas add a figure if possible.

Figures:

Figure 5 is not good for publication in a scientific paper! In the figure the number are too large and the number 3 das not indicate noting. Please use the arrows to indicate the diverse structures.

 Pleas add figures in the results section showing your samples and showing the structures described in the text.

 Discussion section:

In the discussion section, the results should be discussed also in the contest of the rostral epidural rete mirabile function (cooling of the brain, protection of the brain to the thermal stress...etc).

The author did not discuss in the paragraph Discussion the significant differences they found among the species considered in this study. These important parts should be considered in the text.

Line 215, what do you mean? “During the analysis of the anatomy of the rostral epidural rete mirabile on the cross-section, it can be seen that the structure of the rete mirabile varies. Please avoid the terms “…the rete mirabile varies.”

Round 2

Reviewer 2 Report

The manuscript was improved significantly. Congratulations.

Reviewer 3 Report

The manuscript has been sufficiently improved to warrant publication in Animals.